# Observing anyonization of bosons in a quantum gas

Sudipta Dhar[1,6], Botao Wang[2,3,6], Milena Horvath[1,6], Amit Vashisht[2,3], Yi Zeng[1], Mikhail B. Zvonarev[4], Nathan Goldman[2,3,5], Yanliang Guo[1✉], Manuele Landini[1✉] & Hanns-Christoph Nägerl[1✉]

Anyons[1,2] are low-dimensional quasiparticles that obey fractional statistics, hence interpolating between bosons and fermions. In two dimensions, they exist as elementary excitations of fractional quantum Hall states[3–5] and are believed to enable topological quantum computing[6,7]. One-dimensional anyons have been theoretically proposed, but their experimental realization has proven to be difficult. Here we observed emergent anyonic correlations in a one-dimensional strongly interacting quantum gas, resulting from the phenomenon of spin–charge separation[8–10]. A mobile impurity provides the necessary spin degree of freedom to engineer anyonic correlations in the charge sector and simultaneously acts as a probe to reveal these correlations. Starting with bosons, we tune the statistical phase to transmute bosons through anyons to fermions and observe an asymmetric momentum distribution[11–14], a hallmark of anyonic correlations. Going beyond equilibrium conditions, we observed dynamical fermionization of the anyons[15]. This study opens the door to the exploration of non-equilibrium anyonic phenomena in a highly controllable setting[15–17].

Quantum theory tells us that particles can be categorized into two distinct groups on the basis of phase $\theta$ that the quantum wavefunction accumulates when two particles are exchanged[18]. This phase is crucial to the collective behaviour of ensembles of identical particles: bosonic particles, with $\theta = 0$, may pile up and condense into the same state, whereas fermions, with $\theta = \pi$, follow Pauli's exclusion principle and avoid each other. This has drastic consequences, such as forming the basis for the table of elements, assuring the stability of neutron stars in the case of fermions and giving rise to spectacular phenomena (such as superfluidity, superconductivity and laser emission) for bosons. However, in dimensions lower than three, more exotic possibilities exist. In the seminal studies by Leinaas and Myrheim[1] and Wilczek[2], it was realized that a new type of particle, called anyon, with arbitrary values of $\theta$ is possible. Anyons behave as neither bosons nor fermions. They obey fractional quantum statistics[19,20] and are expected to show an intermediate correlation behaviour, interpolating between bosons and fermions.

Two-dimensional (2D) anyons are found to exist as quasiparticles in topological states of matter, such as fractional quantum Hall states in solid-state systems[21,22], and they can be engineered in superconducting quantum processors[23,24] and trapped-ion processors[25]. Triggered by Haldane's fractional exclusion principle[19], which applies to arbitrary spatial dimensions, the existence of one-dimensional (1D) anyons was revealed in the context of the Haldane–Shastry model[26], in which spinon excitations were shown to exhibit fractional statistics. Since then, 1D anyons have attracted a lot of theoretical attention. Numerous phenomena have been proposed, such as statistically induced phase transitions and fractional Mott insulators[27], anomalously bound

pairs[28], accumulation of Friedel oscillations with increasing $\theta$ (ref. 29) and dynamical fermionization and bosonization of anyons[15,30]. Anyonic models in 1D have been studied both in the continuum[31,32] and on discrete lattices[27,33]. As a hallmark for the presence of anyonic correlations, an asymmetric momentum distribution[13,14,34] is expected. The theoretical underpinnings of 1D anyons have long intrigued the scientific community; however, their experimental realization and observation of their dynamical behaviour have remained elusive. A topological gauge theory, developed to describe 1D anyons, was realized using a weakly interacting Bose–Einstein condensate (BEC)[35,36]. Recently, using a Floquet drive, 1D anyons have been realized in a two-atom lattice setting[37].

Here we present a cold-atom realization of a many-body system with anyonic correlations in 1D. We used a degenerate gas of strongly interacting Cs atoms to simulate 1D hardcore anyons with an arbitrary statistical phase $\theta$. Our system consists of a single spin impurity embedded in and strongly interacting with a Tonks–Girardeau[38,39] host gas. The impurity served a dual purpose in our study. It enabled the generation of anyonic correlations in the system and acted as a probe to observe these correlations.

For a strong impurity–host interaction, spin–charge separation occurs in our system[8] (Fig. 1a), with the many-body wavefunction of $N$ particles factorizing into a spatial part $\varphi(x_1, x_2, ..., x_N)$ and a spin part $\chi(\sigma_1, \sigma_2, ..., \sigma_N)$, where $x_i$ is the position and $\sigma_i = \uparrow, \downarrow$ is the spin of the $i$th particle. Anyonic correlations on $\varphi$ arise from engineering a spin wavefunction with fractional exchange symmetry when we restrict to cyclic permutations. Intuitively, if $\chi$ is chosen such that a permutation results in a phase shift of $-\theta$, the spatial part $\varphi$ is forced to have the opposite shift in such a way that the combined wavefunction is still bosonic.

[1]Institut für Experimentalphysik und Zentrum für Quantenphysik, Universität Innsbruck, Innsbruck, Austria. [2]Center for Nonlinear Phenomena and Complex Systems, Université Libre de Bruxelles, Brussels, Belgium. [3]International Solvay Institutes, Brussels, Belgium. [4]Université Paris-Saclay, CNRS, LPTMS, Orsay, France. [5]Laboratoire Kastler Brossel, Collège de France, CNRS, ENS-Université PSL, Sorbonne Université, Paris, France. [6]These authors contributed equally: Sudipta Dhar, Botao Wang, Milena Horvath. ✉e-mail: yanliang.guo@uibk.ac.at; manuele.landini@uibk.ac.at; christoph.naegerl@uibk.ac.at

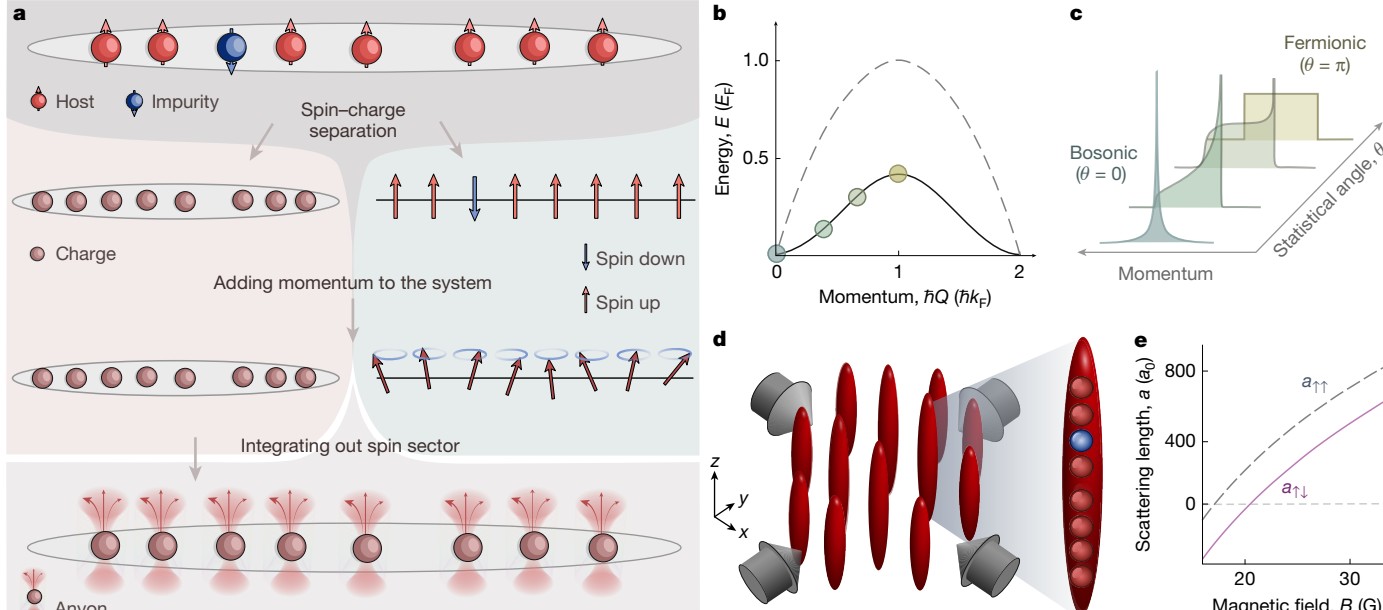

**Fig. 1 | Experimental realization of 1D anyons. a**, Emergence of anyons from spin–charge separation. For strong interactions in one dimension, the wavefunction factorizes into a charge part and a spin part. Note that, although the illustration depicts a localized spin impurity, we created a delocalized impurity. The charge sector (left) depicts one possible positional arrangement of the particles, whereas the spin sector (right) is one possible spin distribution in the squeezed space[40], both corresponding to the arrangement of the spinful particles in the top line. In the finite momentum ground state of the system, all the momentum is carried by the spin sector in the form of spin waves. Integrating out the spin degrees of freedom realizes an effective system of 1D hardcore anyons in the charge sector. The statistical phase $\theta$ of these emerging anyons is given by the momentum of the spin waves (see 'Exchange symmetry engineering' and 'Emergence of anyons via spin–charge separation' in Methods). **b**, Edge of the excitation spectrum of a 1D Bose gas for charge excitation (dashed line) and spin excitation (solid line)[59]. **c**, Expected momentum distributions of anyons[44] for different values of the statistical phase $\theta$ as set by the momentum $\hbar Q$ and indicated in **b**. **d**, Experimental realization of an ensemble of 1D Bose gases in tubes formed by two retro-reflected laser beams. On average, each tube contained one impurity particle (blue sphere). The interaction between the impurity and the Tonks–Girardeau host gas (red spheres) can be tuned by means of a Feshbach resonance[59]. **e**, Host–host (dashed curve) and host–impurity (solid curve) scattering lengths $a_{\uparrow\uparrow}$ and $a_{\uparrow\downarrow}$ as a function of the magnetic field $B$.

For this, the spin wavefunction is prepared in eigenstates of the cyclic spin permutation operator $\hat{C}$, that is, spin waves $|\theta\rangle$ with eigenvalue $e^{-i\theta}$ (see 'Exchange symmetry engineering' in Methods). We experimentally prepared the spin wave by adiabatically accelerating the impurity along the low-energy edge of the excitation spectrum to momentum $\hbar Q$ (Fig. 1b). In each particular state, the momentum of the spin wave fixed the effective phase shift resulting from an exchange of the impurity with one of the particles in the host gas. For $Q = 0$, the exchange resulted in no phase shift, akin to bosonic statistics, whereas for $Q = k_F$, the resulting phase shift was $\pi$ as expected from fermionic statistics (Supplementary Information). For intermediate momenta, we expected that anyonic statistics would be realized. Here $k_F = \rho\pi$ denotes the Fermi momentum of the Tonks–Girardeau gas, with $\rho$ as the 1D density.

A particular observable that is sensitive to the anyonic correlations and to the statistical phase $\theta$ is the momentum distribution of the impurity. Specifically, the one-body correlators of the impurity and for a hardcore anyon system are equal[40,41]:

$$(\langle\varphi|\otimes\langle\theta|)\hat{b}_\downarrow^\dagger(x)\hat{b}_\downarrow(y)(|\theta\rangle\otimes|\varphi\rangle) = \frac{1}{N}\langle\varphi|\hat{a}^\dagger(x)\hat{a}(y)|\varphi\rangle, \qquad (1)$$

where $\hat{b}_\downarrow^\dagger$ ($\hat{b}_\downarrow$) is the bosonic creation (annihilation) operator of the impurity, and $\hat{a}^\dagger$ ($\hat{a}$) is the anyonic creation (annihilation) operator, with $(\hat{a}^\dagger)^2 = 0$ defining the hardcore condition. Equation (1) gives us direct access to the anyonic momentum distribution. Figure 1c illustrates the expected anyonic momentum distribution. As $\theta$ varied, the evolution from bosonic through skewed to fermionic distribution can be clearly seen.

In the experiment, we prepared an array of about 6,000 vertically oriented 1D Bose gases with a weighted average of 37(2) atoms by loading a weakly interacting three-dimensional (3D) BEC of $^{133}$Cs atoms[42] into a 2D optical lattice, as illustrated in Fig. 1d. Initially, all the atoms were in the hyperfine state $|F, m_F\rangle = |3, 3\rangle$, denoted by $|\uparrow\rangle$. A magnetic force levitated the atoms against gravity. We then tuned the 1D interaction strength $g_\uparrow \propto a_{\uparrow\uparrow}$ by means of a Feshbach resonance for the scattering length $a_{\uparrow\uparrow}$ (Fig. 1e) to bring the 1D Bose gases into the Tonks–Girardeau regime, setting the Lieb–Liniger parameter to $\gamma_{\uparrow\uparrow} \approx 14$ (see 'Experiment' in Methods). A short radio-frequency pulse generated spin impurities in $|3, 2\rangle \equiv |\downarrow\rangle$ out of the host gas. On average, we created one impurity per 1D Bose gas, with the number set by the power and duration of the pulse. We transitioned from pure magnetic levitation to a combination of magnetic and optical levitation to allow for a comparatively small force of $F_\downarrow \approx mg/18$ on the impurities, whereas the host gas remained fully levitated. Here $m$ is the mass of the $^{133}$Cs atoms, and $g$ is the gravitational acceleration. The force must be kept small to ensure that the impurity adiabatically follows the lower edge of the excitation spectrum. During the evolution, the impurity experienced a host–impurity interaction strength of $\gamma_{\uparrow\downarrow} \approx 9$, as set by the host–impurity scattering length $a_{\uparrow\downarrow}$ (see Fig. 1e and 'Experiment' in Methods). Applying $F_\downarrow$ for a variable evolution time $\tau$ placed the system at momentum $\hbar Q = F_\downarrow\tau$. Varying $\tau$ from 0 to 3.5 ms set the phase $\theta = Q/\rho = \pi Q/k_F$ to values between 0 and $\pi$. The inhomogeneities of our system affected the values of $\theta$ that we were able to prepare. We always indicated the mean value of the distribution, which we controlled to a few percentage levels. The distribution of densities and therefore $\theta$ values had an estimated root mean square spread of about 20% across the system. Finally, we measured the $|\downarrow\rangle$-momentum distribution $n_\downarrow(k)$ by switching $\gamma_{\uparrow\downarrow}$ to zero and imaging the $|\downarrow\rangle$ atoms after the Stern–Gerlach separation and free time-of-flight expansion. The results are presented in Fig. 2a. For $\theta = 0$, the impurity exhibited a momentum distribution $n_\downarrow(k)$ that

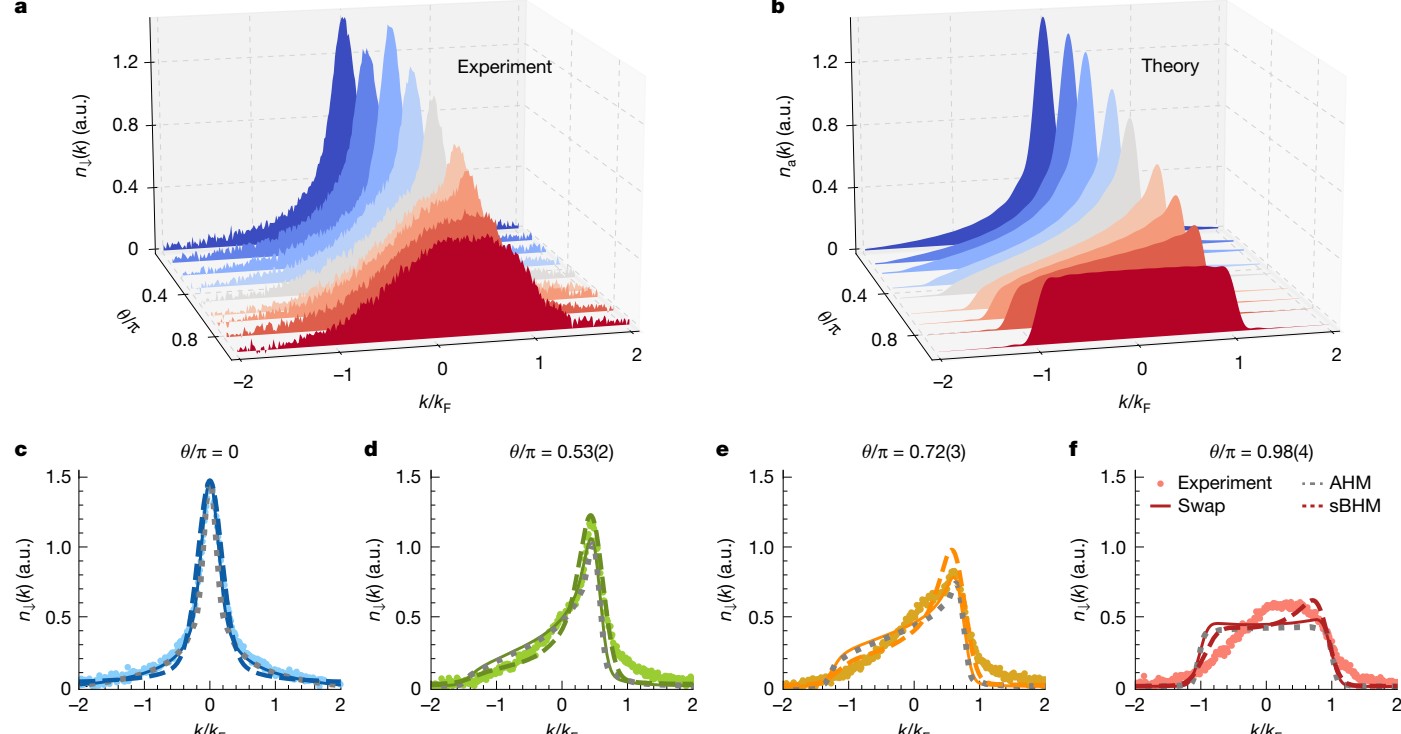

**Fig. 2 | Momentum distribution of anyonized bosons. a**, Evolution of the measured impurity momentum distribution $n_\downarrow(k)$ for variable statistical phases $\theta$ determined by the injected momentum $\hbar Q$, as indicated. Each distribution is the average of seven experimental realizations. **b**, Numerical results of the anyonic momentum distribution $n_a(k)$ using AHM. **c**–**f**, Example distributions for $\theta/\pi$ equal to 0 (**c**), 0.53(2) (**d**), 0.72(3) (**e**) and 0.98(4) (**f**). The error bars are smaller than the symbol sizes. The data were compared to the numerical results of the ground states of AHM (dotted lines), swap model (solid lines) and time evolution governed by the sBHM (dashed lines).

was symmetric and peaked sharply at momentum $\hbar k = 0$. As $\theta$ increased towards $\pi$, the distribution was skewed and the peak gradually disappeared as the distribution broadened and flattened. At $\theta = \pi$, the distribution was flat-top, nearly filling the entire Brillouin zone from $-k_F$ to $k_F$. In essence, the distribution evolved from bosonic to fermionic with significant skewness in between.

The anyonic nature of the skewness behaviour was confirmed by performing a quantitative analysis using several theoretical models. Our system is naturally described by a spinful Lieb–Liniger gas, for which an exact Bethe ansatz solution is available[43] in the limit of a fermionized host gas and which allows an exact anyonic mapping of the impurity momentum distribution in the thermodynamic and hardcore limit[40,44]. To properly capture finite-size effects and directly compare the theoretical prediction to the data, we used lattice models that we expected to reliably describe the system in the low lattice-filling regime (see 'Anyon Hubbard model', 'Dynamical evolution with sBHM' and 'Swap model' in Methods). First, we turn to the anyon Hubbard model (AHM) governed by the Hamiltonian:

$$\hat{H}_{AHM} = -J \sum_\ell \hat{a}_\ell^\dagger \hat{a}_{\ell+1} + \text{h.c.} + \frac{U}{2} \sum_\ell \hat{n}_\ell (\hat{n}_\ell - 1) \tag{2}$$

in the hardcore limit with the on-site interaction $U \to \infty$. Here $J$ is the tunnelling amplitude between nearest-neighbouring sites, $\hat{a}_\ell$ is the anyonic annihilation operator at site $\ell$ and $\hat{n}_\ell = \hat{a}_\ell^\dagger \hat{a}_\ell$ is the particle number operator. The prediction for the momentum distribution, as calculated by a matrix product state algorithm[45], is shown in Fig. 2b. The transition from a peaked bosonic distribution through a skewed to box-like fermionic distribution can be clearly seen. A direct comparison with our data for selected values of $\theta$ is presented in Fig. 2c–f, and we found a reasonably good agreement. The second model we used was

the spinful Bose–Hubbard model (sBHM), aimed at describing the dynamics of a spinful Bose system when a force is applied. Additionally, we make use of a new model, termed the swap model (see 'Swap model' in Methods), featuring swap interactions with a ground state that takes the form of the spin wave that we are targeting. The predictions from these models for a single-tube realization are included in Fig. 2c–f. Given our finite momentum resolution that results from the inhomogeneous tube distribution (see 'Role of finite force and finite interaction' in Methods) of about $\pm 0.2\hbar k_F$, these predictions agree well with our experimental data.

The anyonic correlations are reflected in the asymmetric momentum distribution, exhibiting a shift in the peak of the momentum distribution and a variation in the peak value. Figure 3a shows a comparison of the observed peak position $k^*$ and peak value $n_\downarrow(k = k^*)$ with the calculated behaviour of an anyonic system as $\theta$ was varied. For small $\theta$, the peak momentum was proportional to $\theta$. The slope of the linear dependence was expected to be proportional to the density $\rho$ of the gas[13]. For large $\theta$ close to $\pi$, $k^*$ sharply decreased because $n_\downarrow(k)$ started to transmute into a fermionic distribution. Simultaneously, the peak occupation $n_\downarrow(k = k^*)$ decreased as $\theta$ increased, as shown in Fig. 3b. These observations agree well with the results of the numerical calculations.

Next, we turn to the dynamical properties of our anyonized system. Specifically, we performed a rapidity measurement[46,47], as has been used recently to study dynamical fermionization in Tonks–Girardeau gases[48]. The rapidities were the integrals of motion in a 1D integrable system and were expected to follow a fermionic distribution in the hardcore limit also for anyons[15]. As before, we prepared the system at momentum $\hbar Q$. We then set the force to $F_\downarrow = 0$, and the atoms were allowed to expand in an approximately flat potential in one dimension for a variable time $t_{1D}$ by partially compensating the longitudinal harmonic confinement by means of a horizontally propagating blue-detuned anti-trapping laser

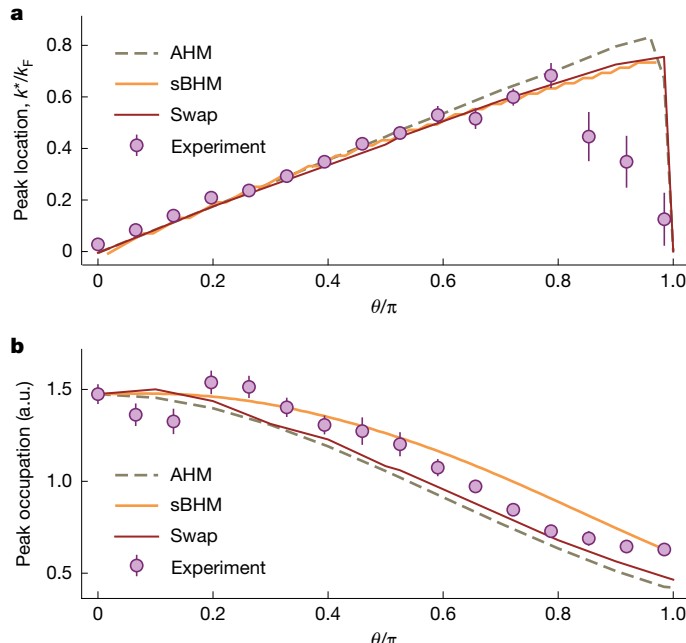

**Fig. 3 | Characterization of the anyonic momentum distribution. a**, Measured peak momentum $k^*$. **b**, Peak occupation of the impurity momentum distribution $n_\downarrow(k)$ as a function of statistical phase $\theta$. The experimental data (dots) were compared with the results of the simulations on the basis of various models, as indicated. The error bars reflect the standard error.

beam. We then measured $n_\downarrow(k)$ as previously described. This procedure mapped the rapidities onto momenta when $t_{1D}$ was chosen to be long enough, typically 5 ms for our parameters, set by the longitudinal trap frequency and the average particle number (see 'Experiment' in Methods). The experimental results are shown in Fig. 4a–c. In the case without 1D expansion ($t_{1D} = 0$), as shown in Fig. 4a, the initial anyonic momentum distributions $n_\downarrow(k)$ for various values of $\theta$ differed greatly

and exhibited the skewness behaviour, as discussed above. However, as $t_{1D}$ was increased, the distributions converged to a similar asymptotic form. At $t_{1D} = 2$ ms (Fig. 4b), the distributions still differed, but at $t_{1D} = 5$ ms (Fig. 4c), they became almost equal. In particular, they have lost their skewness. This behaviour was qualitatively captured by our numerical model, as shown in Fig. 4d,e. We simulated the quench dynamics for $N = 10$ anyons after suddenly releasing the harmonic trapping potential by solving the time-dependent Schrödinger equation (see 'Rapidity of anyons in 1D' in Methods). The evolution from distributions that differ greatly to those that are nearly symmetric and identical can be clearly seen. Note that the expected shape of the distribution in the long-time limit is set by the harmonic trapping potential[15,49]. A box-shaped distribution is expected only for box-shaped trapping. Future experiments with custom-shaped potentials will be able to probe this relationship.

In summary, we have realized a many-body system of 1D anyonized bosons with an arbitrary statistical phase from a strongly interacting spinful bosonic system. Our approach relies on the intrinsic fractionalization of spin and charge degrees of freedom in 1D systems in the presence of strong interactions. The observed asymmetric momentum distributions, a hallmark of anyonic correlations, are in good agreement with the theoretical predictions. Our findings demonstrate the ability to transmute between bosonic and fermionic behaviours by continuously varying the statistical phase, thus creating a flexible system that allows the exploration of anyonic behaviour in a controlled low-dimensional environment. Moreover, the observed phenomenon of dynamical fermionization following a trap quench highlights the complex non-equilibrium dynamics that these systems can exhibit, providing insights into the interplay between quantum statistics and the dynamical properties of 1D anyons.

A promising direction for future research is the realization of tunable interactions between anyons[50,51]. This will open up possibilities for the study of exotic quantum phases[52,53] and phase transitions predicted for 1D anyonic systems[27,54]. Our way of realizing density-dependent statistical angles provides a new opportunity to study intriguing dynamical phenomena owing to the presence of a statistical interface[55]. Local control of the density distribution will be beneficial for precise tuning

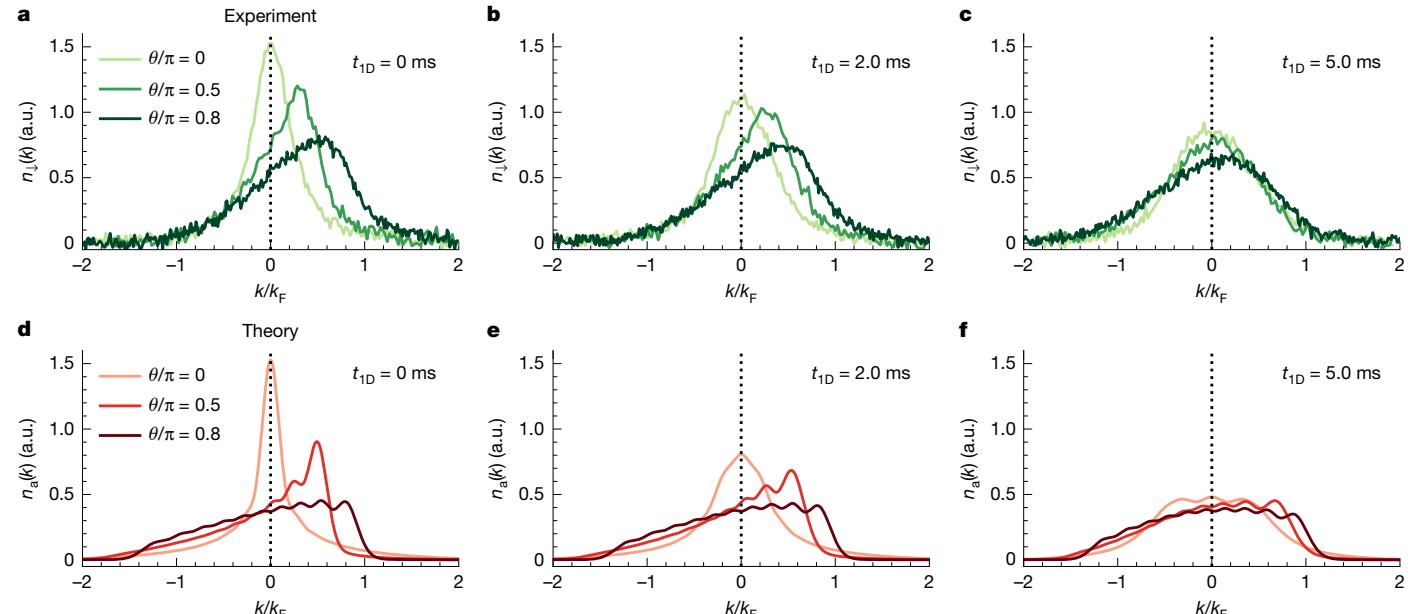

**Fig. 4 | Dynamical fermionization of hardcore anyons. a–c**, Evolution of the impurity momentum distribution $n_\downarrow(k)$ after quenching the confinement to a flat-bottom trap and allowing 1D expansion for $t_{1D} = 0$ ms (**a**), 2 ms (**b**) and 5 ms (**c**) for three different values of statistical phase $\theta$, as indicated. Each distribution is the average of ten experimental realizations. **d–f**, Theoretical prediction for the evolution of the momentum distribution of hardcore anyons ($N = 10$) during 1D free expansion[15] for $t_{1D} = 0$ ms (**d**), 2 ms (**e**) and 5 ms (**f**).

of the statistical phase. Generalizing our study beyond 1D to study topologically non-trivial states of matter is also an interesting avenue. Furthermore, our method of measuring non-local string-type correlators by means of impurities can be used to probe topological order in generic many-body systems[56–58].

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

## Methods

### Experiment

The experiment started with an interaction-tunable 3D BEC of $1.3 \times 10^5$ $^{133}$Cs atoms[60] prepared in the lowest magnetic hyperfine state $|F, m_F\rangle = |3, 3\rangle \equiv |\uparrow\rangle$, held in a crossed-beam dipole trap and levitated against gravity by a magnetic field gradient. The BEC is in the Thomas–Fermi regime with the 3D $s$-wave scattering length $a_{\uparrow\uparrow}$ tuned to $a_{\uparrow\uparrow} \approx 220 \, a_0$, corresponding to an offset magnetic field of $B = 21.24(1)$ G. A 2D optical lattice, generated by two retro-reflected laser beams propagating in orthogonal directions, was gradually ramped up in 500 ms to a potential depth of $30E_r$, with $E_r = \pi^2\hbar^2/(2ma^2)$ the photon recoil energy, cutting the 3D system into an array of 1D tubes that are oriented along the vertical direction, as sketched in Fig. 1d. Here $a = \lambda/2$ is the lattice spacing with $\lambda = 1{,}064.5$ nm the wavelength of the lattice light. The longitudinal trapping frequency in the 1D tubes was 25.6(3) Hz. The magnetic field was then ramped up adiabatically to $B = 35.1$ G, tuning $a_{\uparrow\uparrow}$ to $a_{\uparrow\uparrow} \approx 750 \, a_0$, setting the Lieb–Liniger interaction parameter $\gamma_{\uparrow\uparrow} = mg_\uparrow/(\hbar^2\rho) \approx 14$, where $\rho = N/L \approx 1.33 \, \mu\text{m}^{-1}$ is the average 1D density, and $L$ is the average system length. The nominal value of the Fermi wavevector is given by $k_F = \pi\rho$. Here $g_\uparrow \approx 2\hbar\omega_\perp a_{\uparrow\uparrow}$ (ref. 59), and $\omega_\perp$ is the transversal trap frequency. For these values, the 1D systems are deeply in the fermionized Tonks–Girardeau regime[39,61].

The impurities were Cs atoms that were transferred to the Zeeman substrate $|3, 2\rangle \equiv |\downarrow\rangle$ using a short radio-frequency pulse. Power and duration were set such that, on average, one impurity per tube was created. The pulse duration (15 μs) was much shorter than the Fermi time ($t_F = 120$ μs), ensuring that the spatial profile of the impurity closely matched that of the host gas. The number of impurities varied across the atomic density distribution. Because our detection was sensitive only to the impurity atoms, tubes with no impurities were irrelevant. For tubes with two impurities, the momentum distribution was not expected to be exactly anyonic, but the deviation was small because the impurities were still the minority component. The 3D scattering length between the impurity and host atoms $a_{\uparrow\downarrow}$ also varied with $B$ (Fig. 1e). At $B = 35.1$ G, the host–impurity Lieb–Liniger parameter $\gamma_{\uparrow\downarrow}$ took the value $\gamma_{\uparrow\downarrow} \approx 9$. The impurity atoms in $|\downarrow\rangle$ experienced a smaller levitating force and would be accelerated by $F_\downarrow = mg/3$. Such a comparatively strong force would lead to a non-adiabatic time evolution[59], populating the continuous spectrum of the gapless quantum liquid and pulling the system away from its ground state (see below). To avoid this, we adiabatically turned on optical levitation in 100 ms. Specifically, a 1,064-nm Gaussian beam with a $1/e^2$ waist of $\sigma_z \approx 210$ μm, positioned $\sigma_z/2$ above the atoms, generated a nearly linear optical potential gradient. A laser power of approximately 10 W indiscriminately levitated the host and impurity atoms when the magnetic force was off. A tunable force $F_\downarrow$ on the impurity atoms while still fully levitating the host atoms can then be generated by adjusting the fraction of optical versus magnetic levitation.

### Role of finite force and finite interaction

Here we studied the role of the finite force and finite interaction in our system. In Extended Data Fig. 1a, we show $n_\downarrow(k)$ at a fixed total momentum $\hbar Q \approx \hbar k_F$ for two different values of force $F_\downarrow$. For a strong force $F_\downarrow = mg/3$, the distribution $n_\downarrow(k)$ was skewed and had a peak at around $k = k_F$. By contrast, for a relatively small force $F_\downarrow = mg/18$, the distribution was more symmetric and flat-top, as expected for a fermionic distribution. The simulations on the basis of sBHM were in good agreement with our experimental data. The residual asymmetry in the theoretical curve is attributed to the finite $F_\downarrow$. The deviation between theory and experiment mainly resulted from the inhomogeneities of the experimental system in view of the distribution of $k_F$ values for different tubes. From the observed broadening of the momentum distribution of the system, we estimated an upper bound on the Fermi wavevector variation across the tubes. The root mean square width corresponds to 0.2 $\hbar k_F$, where $k_F$ corresponds to the mean value. Next, we turn to the effect of finite interaction strength on the momentum distribution. In Extended Data Fig. 1b, we showed $n_\downarrow(k)$ at a fixed $Q \approx 0.5k_F$ for three different values of interaction strength $\gamma_{\uparrow\downarrow}$. Close to the non-interacting point $\gamma_{\uparrow\downarrow} \approx 0$, the distribution resembled a bosonic distribution peaked around $k = 0.5k_F$ and did not show any skewness. As we increased the interaction strength to a moderate value of $\gamma_{\uparrow\downarrow} \approx 3$, the height of the peak decreased, and $n_\downarrow(k)$ broadened to the left. Only for a sufficiently strong interaction did the distribution start to agree with the prediction from AHM. This confirmed that strong interactions are crucial for the emergence of anyonic correlations in our system. Note that the peak in the measured $n_\downarrow(k)$ was broader than that in the AHM predictions for a single tube. This is again attributed to the effect of inhomogeneities. Note that $\gamma_{\uparrow\uparrow}$ also varied when $\gamma_{\uparrow\downarrow}$ was changed, but it always stayed above 3.

### Exchange symmetry engineering

We now elaborate on the way in which the emergence of a spin wave in the system led to the appearance of anyonic correlations on the original particles, as expressed by equation (1). Owing to the phenomenon of spin–charge separation, the exchange symmetry of the spatial part is dictated by the exchange symmetry of the spin part of the wavefunction. To obtain an exchange phase of $\theta$ in the spatial wavefunction, we needed to have an exchange phase of $-\theta$ on the spin wavefunction. To describe the system, we used the bosonic version of the approach described in ref. 40, where the spinful bosonic system is replaced by a spinless bosonic charge sector and a spin chain, describing the spin of each atom. The unitary pairwise spin-exchange operators $\hat{\mathcal{E}}_{\ell,\ell'}$ exchange spin $\ell$ with spin $\ell'$ in the spin chain. The set of $\hat{\mathcal{E}}$ operators generates the symmetric group of permutations $S_N$. A fully anyonic wavefunction should be a simultaneous eigenstate of all $\hat{\mathcal{E}}_{\ell,\ell'}$, with the eigenvalue $e^{-i\theta\,\text{sgn}(\ell-\ell')}$.

This state cannot exist for various reasons. Because $\hat{\mathcal{E}}^2$ is the identity operator, the eigenvalues of $\hat{\mathcal{E}}$ are ±1, corresponding to triplet (bosonic) and singlet (fermionic) wavefunctions. Furthermore, two exchange operators of the type $\hat{\mathcal{E}}_{\ell,\ell'}$ and $\hat{\mathcal{E}}_{\ell',\ell''}$ do not commute with each other, as can easily be verified. Simultaneous eigenstates of all pairwise exchange operators are therefore not easy to find, as a result of the fact that the group $S_N$ for $N$ larger than 2 is non-abelian. Nevertheless, certain observables in the form of correlation functions can be sensitive only to a subgroup of exchanges, as shown in the following.

We now try to find the common eigenstates of only a subgroup of $S_N$, with the required form of eigenvalues. In this sense, although this method cannot generate a fully anyonic wavefunction of the host–impurity system, it can at least give us direct access to specific observables of the anyonic gas. We look for a subgroup of $S_N$ with elements that can have complex eigenvalues. The cyclic subgroups are abelian, and the eigenvalues of the different elements are given by the $m$th roots of unity if $m$ is the size of the cycle. We concentrate on the cyclic group of maximal order, $C_N$, because this is the most relevant for us. The group generator $\hat{C}$ performs a cyclic rotation of the spin-chain configuration of the system $\hat{C}|\sigma_1, \ldots, \sigma_N\rangle = |\sigma_N, \sigma_1, \ldots, \sigma_{N-1}\rangle$. The eigenvalues are given by $e^{-i\theta}$, for $\theta = 2\pi n/N$, with $n = 0, \ldots, N-1$, and the eigenstates are spin waves. Let us clarify the connection between the exchange phase and the eigenvalue of $\hat{C}$. One cyclic permutation corresponds to $N-1$ backward binary exchanges. This can be seen by inspecting the effect of the operator on the state of the spin chain. To reproduce the behaviour of anyons with forward exchange phase $-\theta$, the eigenvalue of $\hat{C}$ should correspond to $e^{i\theta(N-1)}$. This reduces to $e^{-i\theta}$ using the condition $\theta N = 2\pi n$, with $n \in \mathbb{Z}$, which is necessary to keep the wavefunction single-valued. The allowed values of $\theta$ are therefore discretized but become dense in the thermodynamic limit. In our experiment, $N \simeq 37$, giving a discretization in steps of $\Delta\theta/\pi \simeq 0.03$, which is below our uncertainty owing to inhomogeneities. In the case of a single impurity, the spin waves take the form

$$|\theta\rangle = \frac{1}{\sqrt{N}} \sum_{\ell=0}^{N-1} e^{i\theta\ell} \hat{C}^\ell |\downarrow, \uparrow, \uparrow \ldots \uparrow\rangle. \qquad (3)$$

We wanted to identify the correlation functions that are well described by the $\hat{C}$ operator. The simplest example is the one-body correlation function of the impurity, for the single-impurity case. To see this connection, consider the action of the operator $\hat{b}_\downarrow^\dagger(x)\hat{b}_\downarrow(y)$ on the spin configuration of the 1D system. The destruction operator is only non-zero if the spin-down particle is found at position $y$, and the creation operator then places it at position $x$. As a result, the spin configuration of the system is shifted by exactly the amount $\hat{N}(x) - \hat{N}(y)$, taking $x > y$. Here $\hat{N}(x) = \int_{-\infty}^{x} \hat{n}(y)dy$ counts the number of particles to the left of $x$. This corresponds to the application of the operator $\hat{C}^{\hat{N}(x)-\hat{N}(y)}$. We can therefore rewrite

$$\hat{b}_\downarrow^\dagger(x)\hat{b}_\downarrow(y) = \hat{b}^\dagger(x)\hat{b}(y)\hat{C}^{\hat{N}(x)-\hat{N}(y)}\hat{\Pi}_\downarrow(\hat{N}(y)), \qquad (4)$$

where $\hat{b}$ is the destruction operator of spinless hardcore bosons in the charge sector, and $\hat{\Pi}_\downarrow(\hat{N}(y))$ is the projector operator on spin down for the spin at position $\hat{N}(y)$ in the spin chain. If the spin state $|\theta\rangle$ is prepared, we get

$$\langle\theta|\hat{b}_\downarrow^\dagger(x)\hat{b}_\downarrow(y)|\theta\rangle = \frac{1}{N} e^{-i\theta\hat{N}(x)} \hat{b}^\dagger(x)\hat{b}(y) e^{i\theta\hat{N}(y)} = \\ = \frac{1}{N}\hat{a}^\dagger(x)\hat{a}(y), \qquad (5)$$

where in the last equivalence we used the Jordan–Wigner transformation $\hat{a} = \hat{b}e^{i\theta\hat{N}}$. The factor $1/N$ results from the mean value of $\hat{\Pi}_\downarrow$ on the spin wave. It is easy to see how this argument can be generalized to the multi-impurity case, giving a family of anyonic correlation functions that can be exactly simulated with this method. Their explicit expression is given by

$$\hat{b}_\sigma^\dagger(x_1) \ldots \hat{b}_\sigma^\dagger(x_m)\hat{b}_\sigma(x_1 + d) \ldots \hat{b}_\sigma(x_m + d) \propto \\ \hat{a}^\dagger(x_1) \ldots \hat{a}^\dagger(x_m)\hat{a}(x_1 + d) \ldots \hat{a}(x_m + d), \qquad (6)$$

where the number $m$ of creation (destruction) operators should match the number of spin $\sigma$ particles in the spin wave. This demonstrates how, whenever the spin-wave state is realized, we can find correlation functions of the original spinful gas that map exactly onto the correlation functions of a system of $N$ anyons, explaining why it is possible to access the momentum distribution of the anyons with measurements on the original spinful bosons. Making use of this equivalence in practice requires control of the spin state of the system, but it is completely independent of the state in the charge sector. It is therefore possible to directly measure the dynamics of the anyonic correlation functions, assuming that the spin wavefunction remains in a spin-wave state during evolution. In our system, we prepared a spin wave as the eigenstate of momentum with the lowest energy by slowly accelerating the impurity.

## Emergence of anyons via spin–charge separation

We now turn to a lattice model to understand how the charge sector can be mapped onto an anyonic gas when the spinful hardcore bosons are prepared in a finite-momentum ground state. We consider the Hamiltonian $\hat{H}_{lat}$ describing a gas of $N$ spinful hardcore bosons:

$$\hat{H}_{lat} = -J \sum_{\ell=1,\sigma}^{L_S-1} \hat{b}_{\sigma\ell}^\dagger \hat{b}_{\sigma\ell+1} - J\sum_\sigma \hat{b}_{\sigma L}^\dagger \hat{b}_{\sigma 1} + \text{h.c.} \qquad (7)$$

Here $\hat{b}_{\sigma\ell}^\dagger$ ($\hat{b}_{\sigma\ell}$) are bosonic creation (annihilation) operators at site $\ell$, $\sigma = (\uparrow, \downarrow)$ is the spin index and $J$ is the hopping amplitude, and its value

is specified below. We assume to be in the low-density limit $N/L_S \ll 1$, where $L_S$ is the number of lattice sites, and we impose periodic boundary conditions so that the conservation of momentum is assured. The operators $\hat{b}_{\sigma\ell}^\dagger$ ($\hat{b}_{\sigma\ell}$) are assumed to satisfy a no-double-occupancy constraint, $\sum_\sigma \hat{b}_{\sigma\ell}^\dagger \hat{b}_{\sigma\ell} \leq 1$. Under this no-double-occupancy constraint, the spin and charge degrees of freedom separate, that is, the wavefunction $|\Psi\rangle$ can be written as $|\Psi\rangle = |\varphi\rangle \otimes |\chi\rangle$. Here $|\varphi\rangle$ and $|\chi\rangle$ denote the wavefunction for the charge and spin parts, respectively. The Hamiltonian $\hat{H}_{lat}$ can be written in spin–charge separated form as[62]

$$\hat{H}_{sc} = -J \sum_{\ell=1}^{L_S-1} \hat{f}_\ell^\dagger \hat{f}_{\ell+1} - J(-1)^{N-1}\hat{f}_{L_S}^\dagger \hat{f}_1 \hat{C}^\dagger + \text{h.c.}, \qquad (8)$$

where $\hat{f}_j^\dagger$ ($\hat{f}_j$) is the spinless fermionic creation (annihilation) operator at site $j$, and $\hat{C}$ is the previously introduced spin permutation operator. Note that a bosonic description of the charge sector, with hardcore constraint, is also possible but has the disadvantage that the bosonic particles are still interacting so that diagonalization is not straightforward. The spin permutation operator $\hat{C}$ and the spinless fermionic operators can be diagonalized separately because they are independent of each other. The eigenstates of $\hat{C}$ are spin waves of the form

$$|\psi_\nu\rangle = \frac{1}{\sqrt{N_\nu}} \sum_{j=0}^{N_\nu-1} e^{i\theta j} \hat{C}^j |\sigma_1, \ldots, \sigma_N\rangle, \qquad (9)$$

where $|\sigma_1, \ldots, \sigma_N\rangle$ is an arbitrary configuration of the spin chain, $\nu$ enumerates all possible disconnected spin blocks and $N_\nu$ corresponds to the total number of distinct elements of the form $\hat{C}^j |\sigma_1, \ldots, \sigma_N\rangle$ in the $\nu$th block. The eigenvalues of $\hat{C}$ are given by $e^{-i\theta}$, for $\theta = 2\pi n/N_\nu$, with $n = 0, \ldots, N_\nu - 1$. In the case of a single impurity $N_\downarrow = 1$, the eigenstates take the form of equation (3). By projecting $\hat{H}_{sc}$ on the eigenspace of $\hat{C}$, we get an effective Hamiltonian for the charge sector

$$\hat{H}_{eff} = -J \sum_{\ell=1}^{L_S-1} \hat{f}_\ell^\dagger \hat{f}_{\ell+1} - J(-1)^{N-1}e^{i\theta}\hat{f}_{L_S}^\dagger \hat{f}_1 + \text{h.c.} \qquad (10)$$

Here we see that the fermionic charge sector acquires an overall flux. This spin-generated flux is a collective effect, imposed by the spin waves onto the charge degrees of freedom. Note that the original Hamiltonian $\hat{H}_{lat}$ does not break time-reversal symmetry. However, time-reversal symmetry is broken for the $\hat{H}_{eff}$ governing the charge sector. This is a result of the projection onto a specific spin-wave subspace. Finally, we performed an anyonic transformation

$$\hat{a}_\ell = \hat{f}_\ell e^{i(\theta+\pi)\hat{N}_\ell} \quad \text{with} \quad \hat{N}_\ell = \sum_{j=1}^{\ell-1} \hat{n}_j \qquad (11)$$

The phase factor in the boundary term vanishes, $(-1)^{N-1}e^{i\theta}e^{i(\theta+\pi)(N-1)} = 1$, and the Hamiltonian $\hat{H}_{eff}$ can be mapped onto a system of hardcore anyons with a periodic boundary condition

$$\hat{H}_{AHM} = -J \sum_{\ell=1}^{L_S-1} \hat{a}_\ell^\dagger \hat{a}_{\ell+1} - J\hat{a}_{L_S}^\dagger \hat{a}_1 + \text{h.c.} \qquad (12)$$

As one can see, the anyonic model does not contain any concatenated flux. The transformation equation (11) is a generalized Jordan–Wigner transformation[63], and the anyons can be understood as composite particles in the charge sector[64,65]. Each spin wave selects a specific value for the statistical phase. In the thermodynamic limit, this result also holds for any choice of boundary conditions. This justifies the use of fixed boundary conditions in the numerics.

Next, we turn to anyonic observables that can be measured experimentally. The real-space density of these anyons can be extracted by measuring the total density of the gas $\langle\varphi|\hat{a}_\ell^\dagger \hat{a}_\ell|\varphi\rangle = \langle\varphi|\hat{f}_\ell^\dagger \hat{f}_\ell|\varphi\rangle = \sum_\sigma \langle\Psi|\hat{b}_{\sigma\ell}^\dagger \hat{b}_{\sigma\ell}|\Psi\rangle$, where $\Psi$ is the many-body wavefunction of the whole

system. However, for hardcore anyons, the real-space density is independent of $\theta$. The one-body correlator $\langle \hat{a}_i^\dagger \hat{a}_j \rangle$, on the other hand, is very sensitive to $\theta$. The Fourier transform of this gives the anyonic momentum distribution, which can be measured by measuring the momentum distribution of the impurity in our system through equation (1). Note that Hamiltonian (equation (10)) can be diagonalized exactly[62]. The momenta of the fermions correspond to the rapidities of the system.

## Anyon Hubbard model

To benchmark the anyonic behaviour realized in the experiment, we next elaborated on the anyonic correlations of the paradigmatic AHM, which can be effectively simulated by using a bosonic model with density-dependent tunnelling. By using a fractional version of the Jordan–Wigner transformation, that is, the anyon–boson mapping

$$\hat{a}_\ell = \hat{b}_\ell e^{i\theta \hat{N}_\ell}, \quad \hat{N}_\ell = \sum_{j=1}^{\ell-1} \hat{n}_j, \tag{13}$$

AHM from equation (2) can be expressed in terms of bosonic operators as

$$\hat{H}_{\text{AHM}}^{\text{B}} = -J \sum_{\ell=1}^{L_S - 1} (\hat{b}_\ell^\dagger \hat{b}_{\ell+1} e^{i\theta \hat{n}_\ell} + \text{h.c.}) + \frac{U}{2} \sum_\ell \hat{n}_\ell (\hat{n}_\ell - 1). \tag{14}$$

Here $\hat{b}_\ell$ are the bosonic annihilation operators at site $\ell$.

Different from the bosonic one-body density correlation $\langle \hat{b}_\ell^\dagger \hat{b}_{\ell'} \rangle$, the correlator of anyons $\langle \hat{a}_\ell^\dagger \hat{a}_{\ell'} \rangle$ can be expressed as

$$\langle \hat{a}_\ell^\dagger \hat{a}_{\ell'} \rangle = \langle \hat{b}_\ell^\dagger e^{i\theta(\hat{N}_{\ell'} - \hat{N}_\ell)} \hat{b}_{\ell'} \rangle. \tag{15}$$

For the data shown in Figs. 2 and 3, we have assumed $N = 10$ anyons in $L_S = 40$ lattice sites in the hardcore limit with open boundary condition. The effect of the boundary condition is negligible for large system sizes. The use of a reduced atom number speeds up considerably the numerics. We found a consistent and satisfactory agreement with the experimental data by considering a large system size at low filling (Supplementary Information).

## Dynamical evolution with sBHM

In practice, a spin wave can be generated by slowly accelerating the impurity. To efficiently simulate such a dynamical process, we considered an sBHM on a 1D lattice:

$$\hat{H}_{\text{sBHM}} = -J \sum_{\ell=1}^{L_S - 1} (\hat{b}_{\uparrow\ell}^\dagger \hat{b}_{\uparrow\ell+1} + \hat{b}_{\downarrow\ell}^\dagger \hat{b}_{\downarrow\ell+1} + \text{h.c.})$$
$$+ U_{\uparrow\downarrow} \sum_\ell \hat{n}_{\uparrow\ell} \hat{n}_{\downarrow\ell} - \sum_\ell F_\downarrow a\ell \hat{n}_{\downarrow\ell}. \tag{16}$$

Here $\hat{b}_{\uparrow\ell}$ and $\hat{b}_{\downarrow\ell}$ are the annihilation operators of the host particles and an impurity at site $\ell$, respectively, with their hopping strength being denoted by $J$. We consider the hardcore limit of the intracomponent interaction, that is, $U_{\uparrow\uparrow} \to \infty$ and $U_{\downarrow\downarrow} \to \infty$. The on-site interaction between the host particles and the impurity is denoted by $U_{\uparrow\downarrow}$. A constant force $F_\downarrow$ is applied only to the impurity. We define the dimensionless force $\mathcal{F} = \frac{F_\downarrow m}{\hbar^2 \rho^3}$. At the low filling limit, any lattice model reduces to a continuum model with the effective mass given by

$$m = \frac{\hbar^2}{2Ja^2}. \tag{17}$$

By setting the value of the effective mass to be equal to the particle's mass, we fix the value of $Ja^2$. By defining the filling factor in a lattice $n = N/L_S$ and $a = L/L_S$ being the lattice constant, one obtains the following mapping between quantities:

$$\frac{U}{J} = \frac{g_{\uparrow\downarrow}}{a} \frac{2ma^2}{\hbar^2} = 2\gamma_{\uparrow\downarrow} \frac{N}{L_S}, \tag{18}$$

$$\frac{F_\downarrow a}{J} = F_\downarrow a \frac{2ma^2}{\hbar^2} = 2\mathcal{F} \left( \frac{N}{L_S} \right)^3. \tag{19}$$

In our simulation, the initial impurity distribution was defined by the ground state of the Hamiltonian (equation (16)) with $F_\downarrow = 0$ and a harmonic trapping potential $V$ applied only for the impurity. At $t = 0$, we suddenly removed the traps and switched on the constant force $F_\downarrow$. We simulated the quench dynamics by solving the time-dependent Schrödinger equation associated with the Hamiltonian (equation (16)) by using the time-dependent variational principle on the basis of matrix product states implemented using ITensors[45,66]. The results are presented in Figs. 2 and 3. The parameters chosen were $L_S = 40$, $N_\downarrow = 1$, $N_\uparrow = 20$, $U/J = 9.1$ and $F_\downarrow a/J = 0.15$ for numerical convenience. A more costly simulation by using a larger system size (for example, $L_S = 120$) at lower filling (for example, $N_\uparrow/L_S = 0.25$) gives very similar results (Supplementary Information).

## Swap model

Inspired by the central role of the spin wave in the emergence of anyonic behaviour of our system, we developed a toy model with a ground state that encodes the spin wave we are targeting:

$$\hat{H}_{\text{swap}} = -J \sum_{\ell=1}^{L_S - 1} \hat{b}_{\uparrow\ell}^\dagger \hat{b}_{\uparrow\ell+1} - J \sum_{\ell=1}^{L_S - 1} \hat{b}_{\downarrow\ell}^\dagger \hat{b}_{\downarrow\ell+1}$$
$$- J_{\text{ex}} e^{i\theta} \sum_{\ell=1}^{L_S - 1} \hat{b}_{\uparrow\ell}^\dagger \hat{b}_{\downarrow\ell+1}^\dagger \hat{b}_{\downarrow\ell} \hat{b}_{\uparrow\ell+1} + \text{h.c.}, \tag{20}$$

with $\hat{b}_{\uparrow\ell}$ and $\hat{b}_{\downarrow\ell}$ being the annihilation operators of the host particles and the impurity at site $\ell$, respectively, and their hopping strength is denoted by $J$. In the strongly interacting regime, the swapping strength $J_{\text{ex}}$ is expected to be of the order of $J^2/U_{\uparrow\downarrow}$. We encoded the spin-wave information by assigning the factor $e^{i\theta}$ to the swapping terms. The ground state of the swap model is expected to effectively describe the lowest energy state of the spinful system for momentum $\hbar Q = \hbar \rho \theta$ (ref. 67).

In a spin–charge separated representation, the one-body correlation function $\langle \hat{b}_{\downarrow\ell}^\dagger \hat{b}_{\downarrow\ell'} \rangle$ of the single impurity can be implemented by hopping of spinless particles and swapping $\hat{\mathcal{E}}$ on the spin chain[40,68]. Taking $\ell' \geq \ell$ as an example, we have

$$\langle \hat{b}_{\downarrow\ell}^\dagger \hat{b}_{\downarrow\ell'} \rangle = \sum_{m',m} \langle \varphi | \hat{b}_\ell^\dagger \hat{b}_{\ell'} \delta_{m,\hat{N}_\ell} \delta_{m',\hat{N}_{\ell'}} | \varphi \rangle$$
$$\times \langle \chi | \hat{\mathcal{E}}_{m,m+1} \cdots \hat{\mathcal{E}}_{m'-1,m'} | \chi \rangle. \tag{21}$$

Here the Kronecker $\delta$ operators ensure that the $\ell'$th site is occupied by the $m'$th spin, and after hopping, the $\ell$th site is occupied by the $m$th spin. In the case of a single impurity, the product of swap operators is related to the $\hat{C}$ operator, as shown previously, giving rise to a spin wave, which leads to the one-body correlator of the impurity shown in equation (1). Calculating the Fourier transform of the one-body correlator of the impurity and using the parameters $L_S = 120$, $N_\downarrow = 1$, $N_\uparrow = 30$ and $J_{\text{ex}}/J = 0.01$, we obtained the quasi-momentum distribution of the impurity shown in Figs. 2 and 3. Note that the small value of swapping strength $J_{\text{ex}}$ is related to the strong host–impurity interaction, and the agreement with experimental data is found for a wide parameter regime (Supplementary Information).

## Rapidity of anyons in one dimension

In Fig. 4d–f, we present the results of the simulation of the quench dynamics of anyonic gases after suddenly removing the harmonic

trap in one dimension. The momentum distribution as a function of evolution time is expressed as

$$n_a(k, t) = \frac{1}{2\pi} \iint dx\,dy\, e^{ik(x-y)} \rho_{HCA}(x, y; t), \qquad (22)$$

with the single-particle density matrix of hardcore anyons $\rho_{HCA}(x, y; t)$. Following ref. 15, it can be efficiently computed as

$$\rho_{HCA}(x, y; t) = \sum_{m,n=0}^{N-1} \phi_m^*(x, t) A_{mn}(x, y; t) \phi_n(y, t), \qquad (23)$$

where $A_{mn}(x, y; t)$ are the matrix elements of $\mathbf{A}(x, y; t) = (\mathbf{P}^{-1})^T \det\mathbf{P}$, and the elements of matrix $\mathbf{P}(x, y; t)$ are $P_{mn}(x, y; t) = \delta_{mn} - (1 - e^{-i\theta\operatorname{sgn}(y-x)})\operatorname{sgn}(y-x)\int_x^y dz\,\phi_m^*(z, t)\phi_n(z, t)$. Here, $\phi_n(x, 0)$ are the single-particle wavefunctions of the 1D harmonic oscillator, and $\phi_n(x, t)$ fulfill the time-dependent Schrödinger equation

$$i\hbar \frac{\partial \phi_n(x, t)}{\partial t} = \left( -\frac{\hbar^2}{2m} \frac{\partial^2}{\partial x^2} + \frac{m\omega_0^2 x^2 \Theta(-t)}{2} \right) \phi_n(x, t), \qquad (24)$$

with Heaviside step function $\Theta(t)$, which models a sudden quench $\omega(t) = \omega_0 \Theta(-t)$. The solution was found to be $\phi_n(x, t) = \phi_n(x/b(t), 0) e^{imx^2 \dot{b}/2b\hbar - iE_n\tau(t)/\hbar}/\sqrt{b(t)}$, with the scaling factor $b(t) = \sqrt{1 + \omega_0^2 t^2}$, $\tau(t) = \int_0^t dt'/b^2(t')$ and $E_n = \hbar\omega_0(n + 1/2)$. In the experiment, the trapping frequency was set to $\omega_0 = 2\pi \times 25.6(3)$ Hz, and the average Fermi time was $t_F = 2m/\hbar k_F^2 \approx 0.12$ ms. Owing to the finite size of the optical levitation beam, the expansion time $t_{1D}$ was limited to about 5 ms in the experiment.

## Data availability

The data shown in the main text are available at Zenodo (https://doi.org/10.5281/zenodo.14567996)[69].

## Code availability

Codes that support the findings of this study are available from the corresponding authors upon reasonable request.

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

**Acknowledgements** We thank P. Zechmann and M. Knap for discussions and for providing us at the early stage of this study with the results of simulations to understand the role of a finite force. We thank X.-W. Guan for insightful discussion about the fractional statistics of an impurity immersed in spin-polarized Fermi gases. The Innsbruck team acknowledges funding by a Wittgenstein prize grant under the Austrian Science Fund (FWF) project number Z336-N36 by the European Research Council under project number 789017, by an FFG infrastructure grant with project number FO999896041 and by the FWF's COE 1 and quantA. M.H. thanks the doctoral school ALM for hospitality, with funding from FWF under project number W1259-N27. Work in Brussels was supported by the European Research Council (LATIS project), the EOS project CHEQS, the FRS–FNRS Belgium and the Fondation ULB. Computational resources were provided by the Consortium des Équipements de Calcul Intensif (CÉCI), funded by the Fonds de la Recherche Scientifique de Belgique (FRS–FNRS) under grant no. 2.5020.11 and by the Walloon Region. This research was funded in part by the Austrian Science Fund (FWF) 10.55776/Z336. For open access purposes, the authors have applied a CC BY public copyright license to any author accepted manuscript version arising from this submission.

**Author contributions** This study was conceived by S.D., M.B.Z., M.L. and H.-C.N. Experiments were prepared and performed by S.D. and M.H. Data were analysed by S.D., B.W. and Y.G. Numerical simulations were performed by B.W. and A.V. The theoretical models were developed by M.L., S.D., M.B.Z., B.W., A.V. and N.G. The paper was drafted by H.-C.N., M.L., S.D., B.W., Y.G., M.H., M.B.Z. and Y.Z. All authors contributed to the discussion and finalization of the paper.

**Funding** Open access funding provided by University of Innsbruck and Medical University of Innsbruck.

**Competing interests** The authors declare no competing interests.

**Additional information**
**Correspondence and requests for materials** should be addressed to Yanliang Guo, Manuele Landini or Hanns-Christoph Nägerl.

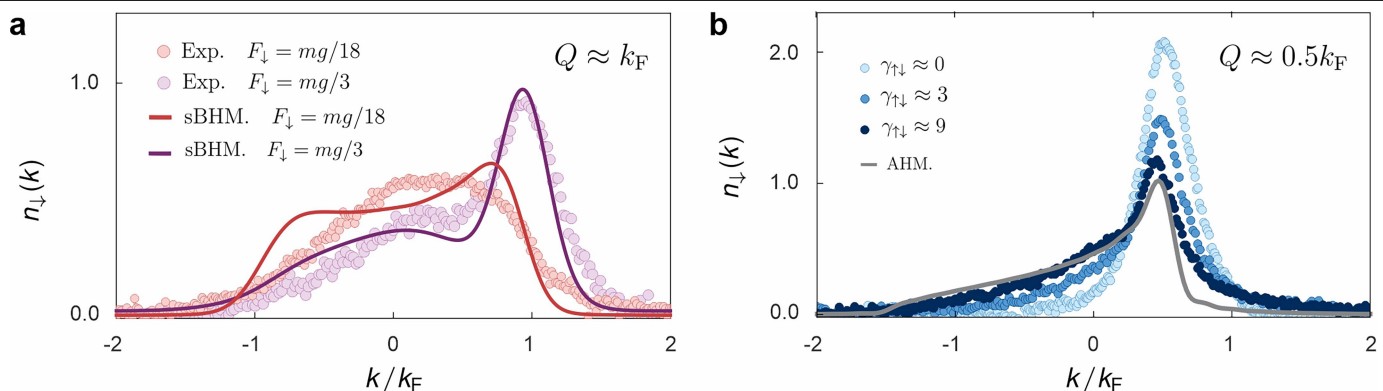

**Extended Data Fig. 1 | Role of finite force and finite interaction. a** Measured $n_\downarrow(k)$ at fixed total momentum $Q$ and fixed interaction strength $\gamma_{\uparrow\downarrow}$ for two different values of the force $F_\downarrow$ as indicated. Each distribution is the average of 7 experimental realizations. The experimental data is compared to the results of the simulations based on the sBHM. **b** Measured $n_\downarrow(k)$ at fixed $Q$ for different $\gamma_{\uparrow\downarrow}$. The solid line is the prediction from AHM.