## [Peer Review File · Nature]

Observing anyonization of bosons in a quantum gas

Corresponding Author: Professor Hanns-Christoph Nägerl

Version 0:

Reviewer comments:

Referee #1

(Remarks to the Author)

Following the Nature review guidelines the following points are highlighted with the report of the submitted manuscript.

A Summary of the key results

The manuscript reports about a breakthrough experiment, which simulates 1D hardcore anyons with an arbitrary statistical phase in a degenerate gas of strongly interacting Cs atoms. The underlying physical idea is to use an impurity, which enables the generation of anyonic correlations in the system and acts at the same time as a probe to observe these correlations. A first key result is the evolution of the measured momentum distribution of anyonized bosons in Fig. 2 for different statistical phases, which are tunable by the injected momentum upon the impurity. This is insofar remarkable as it was a common belief in the literature that such asymmetric anyonic correlations, which are characterized by a peak momentum and occupation depicted separately in Fig. 3, could not directly be realized. A second key result is the dynamical fermionization of hardcore anyons observed in Fig. 4 after quenching the confinement to a flat-bottom trap and allowing a 1D expansion. All these experimental results are complemented by a detailed modelling and the elaboration of a corresponding theoretical framework.

B Originality and significance: if not novel, please include reference

The work is original as it picks up the theoretical proposals [47,52] and realizes them as described in A. In contrast to the first realization of 1D anyons last year in Ref. [45], which is based on a Floquet scheme, the present work manages to directly measure both static and dynamic anyonic correlations. This is not only a unique but also a difficult experiment to perform, which is based on the long-term experience of the experimental team.

Furthermore, the work is also significant in the following two aspects:

1) Since decades it was a goal to determine how to interpolate between the Bose-Einstein and the Fermi-Dirac statistics in form of an anyonic statistics. All previous suggestions for such a fractional statistics proposed in the literature were shown not to apply for anyons via a virial expansion, see for instance

A. Khare, Fractional Statistics and Quantum Theory, 2nd Edn. (World Scientific, 1990)

The underlying failure to find this anyonic statistics is based on the fact that a fractional statistics implies a statistical interaction, which is not exactly describable. Therefore, the only remaining practical approaches are to approximate this fractional statistics analytically or to determine it numerically, as was performed numerous times in the literature, or to measure it. And exactly the latter approach was now successfully realized for the first time within the present work.

2) The results promise a plethora of future research directions for studying statistically induced quantum phase transitions and for investigating dynamical phenomena. In this context it is of importance that the statistical angle is tunable by both the variable evolution time during the applied force and the cloud density. This offers exciting possibilities for unraveling physical effects which involve also spatial and/or temporal changes of the statistical angle.

C Data & methodology: validity of approach, quality of data, quality of presentation

There are the following points, where the quality of data and their presentation could be improved:

- 1) The used Cesium isotope is not mentioned.
- 2) The abbreviation L is used for two different quantities. On page 11 L denotes the number of sites, and on page 8 L denotes the length in one tube.
- 3) Within the main body of the article one should refer precisely to the methods section and vice versa.
- 4) The caption of Fig. 2 should be improved along the following lines:
 - a) The method section E mentions at page 11 that $N=10$, $L=40$ are involved. This contradicts the average atom number $N=37$ mentioned on page 2.
 - b) According to page 9 the statistical angle is, in principle, not continuous but is given by $2\pi n/N$. Therefore, it would be appropriate to mention not only the particle number N but also the integer n which leads to the statistical angles used in subfigures d, e, f.
- 5) The manuscript should clarify the boundary conditions for the experiment and the theoretical modelling. It is inconsistent to use periodic boundary conditions in (10) but not in (14).
- 6) The paper relies on the hard core limit. Therefore, it would be justified to comment up to which extent it might be possible that the experimental realization might be extended to finite two-particle interactions.

D Appropriate use of statistics and treatment of uncertainties

The experimental data involve the following uncertainties concerning the precise value of the statistical angle:

- 1) On average one impurity is in one tube. But also zero or two impurities could be in some tubes.
- 2) The overall harmonic trap along the tube and a variation of the particle number from tube to tube modulate the density and thus affect via the Fermi wave vector the statistical angle.

Therefore at some point the manuscript should briefly comment about the errors involved in the statistical angle.

E Conclusions: robustness, validity, reliability

Here I found no point to comment about.

F Suggested improvements: experiments, data for possible revision

The momentum distribution of Fig. 2F for the pseudo-fermionic case shows a larger discrepancy between theory and experiment. The same seems also to apply for the last three points of Fig. 3a. What is the limitation for not obtaining in the experiment a flat momentum distribution? Could new and more refined measurements improve this?

G References: appropriate credit to previous work?

- 1) An important experiment on the Tonks-Girardeau gas is missing in the list of references:

B. Paredes, A. Widera, V. Murg, O. Mandel, S. Fölling, I. Cirac, G. V. Shlyapnikov, T. W. Hänsch, and I. Bloch, Tonks-Girardeau gas of ultracold atoms in an optical lattice, *Nature* 429, 277 (2004).

This missing reference should even be cited before [48] as it was published a few months earlier.

- 2) There are two references, which promise a third route to anyons and should, therefore, also be cited:

a) C. S. Chisholm, A. Frölian, E. Neri, R. Ramos, L. Tarruell, A. Celi, Encoding a one-dimensional topological gauge theory in a Raman-coupled Bose-Einstein condensate, *Phys. Rev. Research* 4, 043088 (2022)

b) A. Frölian, C. S. Chisholm, E. Neri, C. R. Cabrera, R. Ramos, A. Celi, and L. Tarruell, Realizing a 1D topological gauge theory in an optically dressed BEC, *Nature* 608, 293 (2022)

H Clarity and context: lucidity of abstract/summary, appropriateness of abstract, introduction and conclusions

Here I found no point to comment about.

Referee #2

(Remarks to the Author)

This is a remarkably sophisticated experimental paper addressing an issue that is of broad physical interest. The authors excite a small fraction of atoms in each gas in a bundle of 1D gases and then accelerate the associated spin excitation by a variable amount. They show that each acceleration time is associated with a phase shift of the impurity relative to the host gas, and each phase shift can be associated with exchange statistics. 0 phase is bosonic, π is fermionic, and in between is anyonic. It seems a little like magic, but is still quite convincing, since the signature of the effect (previously theoretically predicted) is an asymmetric momentum distribution for the impurities that is strikingly demonstrated in Fig. 2. They go further in showing dynamical fermionization of hard core anyons, a slightly less convincing but still quite impressive result. I feel that this is a slam dunk Nature paper. The paper is generally clear and well written.

I would not claim to fully understand the sophisticated mappings that underlie these demonstrations, which requires several pages of Methods and supplementary materials. Hopefully a theorist will critique that aspect of the presentation. I do, however, have a couple of suggestions.

On the right side of the cartoon of Fig. 1a, a localized spin flip is shown, although of course the spin flip itself is delocalized. Maybe that part of the figure does not need to be changed, but it might be clearer to note that fact in the caption. That would also make it easier to see why adding momentum creates a spin wave in the system. That part of the figure is clear, but it doesn't really follow (in a cartoon sense) from the previous line. On the left side there is a spatially localized charge hole, in a different place from the spatially localized spin flip. Why isn't it in the same place? Wouldn't the spin charge separation only occur after the spin is accelerated? I can imagine that I am missing some point here, but perhaps the paper could make these points more clearly. In the last line of the figure, and the associated part of the caption, the anyons are created by "integrating out the spin sector". This is the conceptually crucial step in the anyon story and I think it should be made less mysterious at this point in the paper. I realize that there is underlying math in Methods D, but I think the paper would be much more satisfying if more time was devoted at this point in the paper to explaining this transformation to anyons.

On p. 3 where it says "A small force is needed...", it would be clearer to say "The force must be kept small..."

We thank the two reviewers for their constructive comments. In the following, we respond point-by-point.

I. REPLY TO REVIEWER 1

Referee's comment : Following the Nature review guidelines the following points are highlighted with the report of the submitted manuscript.

A. Summary of the key results:

The manuscript reports about a breakthrough experiment, which simulates 1D hardcore anyons with an arbitrary statistical phase in a degenerate gas of strongly interacting Cs atoms. The underlying physical idea is to use an impurity, which enables the generation of anyonic correlations in the system and acts at the same time as a probe to observe these correlations. A first key result is the evolution of the measured momentum distribution of anyonized bosons in Fig. 2 for different statistical phases, which are tunable by the injected momentum upon the impurity. This is insofar remarkable as it was a common belief in the literature that such asymmetric anyonic correlations, which are characterized by a peak momentum and occupation depicted separately in Fig. 3, could not directly be realized. A second key result is the dynamical fermionization of hardcore anyons observed in Fig. 4 after quenching the confinement to a flat-bottom trap and allowing a 1D expansion. All these experimental results are complemented by a detailed modelling and the elaboration of a corresponding theoretical framework.

B. Originality and significance: if not novel, please include reference:

The work is original as it picks up the theoretical proposals [47,52] and realizes them as described in A. In contrast to the first realization of 1D anyons last year in Ref. [45], which is based on a Floquet scheme, the present work manages to directly measure both static and dynamic anyonic correlations. This is not only a unique but also a difficult experiment to perform, which is based on the long-term experience of the experimental team. Furthermore, the work is also significant in the following two aspects:

1) Since decades it was a goal to determine how to interpolate between the Bose-Einstein and the Fermi-Dirac statistics in form of an anyonic statistics. All previous suggestions for such a fractional statistics proposed in the literature were shown not to apply for anyons via a virial expansion, see for instance, "A. Khare, Fractional Statistics and Quantum Theory, 2nd Edn. (World Scientific, 1990)". The underlying failure to find this anyonic statistics is based on the fact that a fractional statistics implies a statistical interaction, which is not exactly describable. Therefore, the only remaining practical approaches are to approximate this fractional statistics analytically or to determine it numerically, as was performed numerous times in the literature, or to measure it. And exactly the latter approach was now successfully realized for the first time within the present work.

2) The results promise a plethora of future research directions for studying statistically induced quantum phase transitions and for investigating dynamical phenomena. In this context it is of importance that the statistical angle is tunable by both the variable evolution time during the applied force and the cloud density. This offers exciting possibilities for unraveling physical effect which involve also spatial and/or temporal changes of the statistical angle.

Response : We sincerely appreciate the reviewer's thorough evaluation of our work and his/her positive recognition of its rigor, originality, and significance.

Referee's comment : C. Data & methodology: validity of approach, quality of data, quality of presentation:

There are the following points, where the quality of data and their presentation could be improved:

1) The used Cesium isotope is not mentioned

Response : We now mention the mass number for the relevant isotope ^{133}Cs .

Referee's comment : 2) The abbreviation L is used for two different quantities. On page 11 L denotes the number of sites, and on page 8 L denotes the length in one tube.

Response : We have introduced the new symbol L_S to denote the number of sites to avoid confusion.

Referee's comment : 3) Within the main body of the article one should refer precisely to the methods section and vice versa.

Response : We don't fully understand the indication by the referee, as we referred previously to the methods in the main article and vice versa. We now indicate precisely each sections in the methods.

Referee's comment : 4) The caption of Fig. 2 should be improved along the following lines:

a) The method section E mentions at page 11 that $N=10$, $L=40$ are involved. This contradicts the average atom number $N=37$ mentioned on page 2.

Response : Choosing a different number is for numerical convenience, as the simulation of anyonic correlation via Jordan-Wigner transformation is time-consuming. In Fig. S1, we have systematically studied the effect of using different numbers of particles and lattice sites. We find a consistent agreement with the experimental data for systems with large size and low filling. Now we have commented on this at the end of Section E in Methods.

Referee's comment : b) According to page 9 the statistical angle is, in principle, not continuous but is given by $2\pi n/N$. Therefore, it would be appropriate to mention not only the particle number N but also the integer n which leads to the statistical angles used in subfigures d, e, f.

Response : We now indicate in page 9 that, in the thermodynamic limit, the θ parameter can take continuous values. In the experiment, the number of atoms is about 37. This corresponds to a discretization of the statistical parameter $\Delta\theta/\pi \simeq 0.03$, which is below our experimental resolution, see also the further discussion about this.

Referee's comment : 5) The manuscript should clarify the boundary conditions for the experiment and the theoretical modelling. It is inconsistent to use periodic boundary conditions in (10) but not in (14).

Response : The physics we are describing strongly relies on conservation of momentum, which is not exactly given for open boundary conditions (OBCs). That is why in theoretical arguments aimed at explaining the main mechanism, we use periodic boundary conditions (PBCs). For some of the simulations, open boundary conditions are numerically more convenient. As momentum conservation becomes increasingly sharp approaching the thermodynamic limit, we have observed that the open-boundary simulations agree more and more with the theoretical picture based on periodic boundary conditions. Although the quasi-momentum is not a good quantum number in a system with open boundary condition, one could still formally apply the Fourier transformation to compute the distribution. With the increase of the system size, the quasi-momentum distribution results obtained from using OBCs approach the case of PBCs (see Figure. 1 below). In the revised manuscript, we comment on the boundary condition at the end of Section E in Methods.

Referee's comment : 6) The paper relies on the hard core limit. Therefore, it would be justified to comment up to which extent it might be possible that the experimental realization might be extended to finite two-particle interactions.

Response : In Fig. 5 and Fig. S4 we have studied the effect of finite interactions, verifying that our

Figure 1: Quasi-momentum distribution of a single impurity in our swap model under periodic (blue dots) and open (orange solid lines) boundary conditions for system size (a) $L_S = 20$ and (b) $L_S = 60$. Other parameters are $N_\uparrow/L_S = 1/2$, $J_{\text{ex}} = 0.1$ and $\theta = \pi/2$.

results are robust in strongly-interacting regime when relaxing the hard-core condition.

Referee's comment : D. Appropriate use of statistics and treatment of uncertainties:

The experimental data involve the following uncertainties concerning the precise value of the statistical angle:

- 1) On average one impurity is in one tube. But also zero or two impurities could be in some tubes.

Response : We now discuss this point in more detail in the methods, section A. Tubes with zero impurities do not contribute to the observed momentum distribution of the spin-down component. Double occupancy can lead to deviations in the experimental signal. The momentum distribution of those tubes is not expected to reproduce exactly the anyonic momentum distribution, but should display a mixture of anyonic character and bosonic one. Intuitively, the fraction of bosonic character should be of the order of $1/(N-2)$, because there are $N-2$ spin-up atoms and only 1 spin-down that the spin-down particle can exchange with. So, if N is large, the effect is expected to be small.

Referee's comment : 2) The overall harmonic trap along the tube and a variation of the particle number from tube to tube modulate the density and thus affect via the Fermi wave vector the statistical angle. Therefore at some point the manuscript should briefly comment about the errors involved in the statistical angle.

Response : We have estimated the spread in k_F values expected in the atomic density distribution. The rms variation of this distribution is 20% of the average value. This corresponds to a similar inhomogeneous broadening on the value of θ across the atomic cloud. What is reported in Fig.2 is the mean of the distribution, with its own error reflecting $\pm 5\%$ uncertainty in the total atom number. We now add a discussion of the broadening of θ in the main text and methods.

Referee's comment : E. Conclusions: robustness, validity, reliability:

Here I found no point to comment about.

F. Suggested improvements: experiments, data for possible revision:

The momentum distribution of Fig. 2F for the pseudo-fermionic case shows a larger discrepancy between theory and experiment. The same seems also to apply for the last three points of Fig. 3a. What is the limitation for not obtaining in the experiment a flat momentum distribution? Could new and more refined measurements improve this?

Response : As now more thoroughly discussed, the main deviations are due to imperfect adiabaticity during preparation and inhomogeneities in the value of θ across the system. Indeed, the use of a box trap, preparation by even weaker forces, or local addressing by means of a quantum-gas microscope will help a lot in the future to reduce such experimental imperfections. In fact, we are gearing up for these improvements. We add a short mention in the outlook section.

Referee's comment : G: References: appropriate credit to previous work?

1) An important experiment on the Tonks-Girardeau gas is missing in the list of references:
B. Paredes, A. Widera, V. Murg, O. Mandel, S. Fölling, I. Cirac, G. V. Shlyapnikov, T. W. Hänsch, and I. Bloch, Tonks-Girardeau gas of ultracold atoms in an optical lattice, *Nature* 429, 277 (2004).

This missing reference should even be cited before [48] as it was published a few months earlier.

2) There are two references, which promise a third route to anyons and should, therefore, also be cited:
a) C. S. Chisholm, A. Frölian, E. Neri, R. Ramos, L. Tarruell, A. Celi, Encoding a one-dimensional topological gauge theory in a Raman-coupled Bose-Einstein condensate, *Phys. Rev. Research* 4, 043088 (2022)

b) A. Frölian, C. S. Chisholm, E. Neri, C. R. Cabrera, R. Ramos, A. Celi, and L. Tarruell, Realizing a 1D topological gauge theory in an optically dressed BEC, *Nature* 608, 293 (2022)

Response : We now add the references indicated by the referee.

Referee's comment : H. Clarity and context: lucidity of abstract/summary, appropriateness of abstract, introduction and conclusions

Here I found no point to comment about.

II. REPLY TO REVIEWER 2

Referee's comment : This is a remarkably sophisticated experimental paper addressing an issue that is of broad physical interest. The authors excite a small fraction of atoms in each gas in a bundle of 1D gases and then accelerate the associated spin excitation by a variable amount. They show that each acceleration time is associated with a phase shift of the impurity relative to the host gas, and each phase shift can be associated with exchange statistics. 0 phase is bosonic, π is fermionic, and in between is anyonic. It seems a little like magic, but is still quite convincing, since the signature of the effect (previously theoretically predicted) is an asymmetric momentum distribution for the impurities that is strikingly demonstrated in Fig. 2. They go further in showing dynamical fermionization of hard core anyons, a slightly less convincing but still quite impressive result. I feel that this is a slam dunk Nature paper. The paper is generally clear and well written.

Response : We sincerely appreciate the reviewer's positive assessment of our work.

Referee's comment : I would not claim to fully understand the sophisticated mappings that underlie these demonstrations, which requires several pages of Methods and supplementary materials. Hopefully a theorist will critique that aspect of the presentation. I do, however, have a couple of suggestions.

On the right side of the cartoon of Fig. 1a, a localized spin flip is shown, although of course the spin flip itself is delocalized. Maybe that part of the figure does not need to be changed, but it might be clearer to note that fact in the caption. That would also make it easier to see why adding momentum creates a spin wave in the system. That part of the figure is clear, but it doesn't really follow (in a cartoon sense) from the previous line. On the left side there is a spatially localized charge hole, in a different place from the spatially localized spin flip. Why isn't it in the same place? Wouldn't the spin charge separation only occur after the spin is accelerated? I can imagine that I am missing some point here, but perhaps the paper could make these points more clearly.

Response : The reviewer is correct that the impurity is created in a delocalized state, but we depict it as localized in our illustration. This is done for simplicity. To clarify the point raised by the reviewer, we have now added a sentence to the figure caption. The difficulty in pictorially representing delocalized particles likely contributed to some misunderstanding, and we have improved the discussion accordingly. The "balls" in the illustration represent possible particle positions rather than exact quantum states. Additionally, the charge hole does not need to align with the position of the impurity but rather with the missing particle in the upper panel. Spin-charge separation is a property of the wavefunction even before the acceleration of the impurity, with spins residing in the "squeezed space" (see ref. 38.)

Referee's comment : In the last line of the figure, and the associated part of the caption, the anyons are created by "integrating out the spin sector". This is the conceptually crucial step in the anyon story and I think it should be made less mysterious at this point in the paper. I realize that there is underlying math in Methods D, but I think the paper would be much more satisfying if more time was devoted at this point in the paper to explaining this transformation to anyons.

Response : As the reviewer points out, we have ample discussion on the exact meaning of the sentence "integrating out the spin sector" in the methods section. We now promote a small part of that explanation in the main text to address the reviewer's suggestion.

Referee's comment : On p. 3 where it says "A small force is needed...", it would be clearer to say "The force must be kept small..."

Response : We have adopted the reviewer's proposed formulation.